# Impact of image averaging on vessel detection using optical coherence tomography angiography in eyes with macular oedema and in healthy eyes

**Hugo Le Boité, Mardoche Chetrit, Ali Erginay, Sophie Bonnin, Carlo Lavia, Ramin Tadayoni, Aude Couturier** *

Ophthalmology Department, Université de Paris, AP-HP, Hôpital Lariboisière, Paris, France

* aude.couturier@aphp.fr

## Abstract

### Purpose

To assess the repeatability of multiple automatic vessel density (VD) measurements and the effect of image averaging on vessel detection by optical coherence tomography angiography (OCTA).

### Methods

An observational study was conducted in a series of healthy volunteers and patients with macular oedema. Five sequential OCTA images were acquired for each eye using the Opto-Vue HD device. The effect of the averaging of the 5 acquisitions on vessel detection was analysed quantitatively using a pixel-by-pixel automated analysis. In addition, two independent retina experts qualitatively assessed the change in vessel detection in averaged images segmented in 9 boxes and compared to the first non-averaged image.

### Results

The automatic VD measurement in OCTA images showed a good repeatability with an overall mean intra-class correlation coefficient (ICC) of 0.924. The mean ICC was higher in healthy eyes compared to eyes with macular oedema (0.877 *versus* 0.960; p < 0.001) and in the superficial vascular plexus *versus* the deep vascular complex (0.967 *versus* 0.888; p = 0.001). The quantitative analysis of the effect of the averaging showed that averaged images had a mean gain of 790.4 pixels/box, located around or completing interruptions in the vessel walls, and a mean loss of 727.2 pixels/box. The qualitative analysis of the averaged images showed that 99.6% of boxes in the averaged images had a gain in vessel detection (i.e., vessels detected in the averaged image but not in the non-averaged image). The loss of pixels was due to a reduction in background noise and motion artifacts in all cases and no case of loss of vessel detection was observed.

**Data Availability Statement:** All relevant data are within the paper and its Supporting information files.

**Funding:** The author(s) received no specific funding for this work.

**Competing interests:** I have read the journal's policy and the authors of this manuscript have the following competing interests: Dr. LE BOITE, Dr. CHETRIT, Dr ERGINAY and Dr LAVIA have nothing to disclose. Dr BONNIN reports personal fees from Allergan, outside of the submitted work. Dr. COUTURIER reports personal fees from Allergan, Bayer, Horus and Novartis, outside of the submitted work. Dr. TADAYONI reports personal fees from Novartis, Bayer, Roche, Genentech, Allergan, Zeiss, Alcon and Oculis, outside of the submitted work. All authors attest that they meet the current ICMJE criteria for authorship.

## Conclusion

The automatic VD measurement using the OptoVue HD device showed a good repeatability in 5 acquisitions in a row setting. Averaging images increased vessel detection, and in about a third of boxes, decreased the background noise, both in healthy eyes and, in a greater proportion, in eyes with macular oedema.

## Introduction

Optical Coherence Tomography Angiography (OCTA) is a dye-free OCT-based imaging technique allowing assessment the retinal vasculature using repeated acquisitions to capture the dynamic motion of erythrocytes [1–3]. It provides a three-dimensional mapping of the different layers of the retinal and choroidal vasculature, including the intermediate and deep retinal vascular plexuses, that cannot be observed using fluorescein angiography [1], and can provide quantitative measurements (for example of the vascular density), with satisfactory repeatability [4–6]. However, single OCTA images often show a significant background noise, discontinuities in vessel limits, and artifacts due to shadow of the superficial vessels (projection artifacts), eye movement or display artifacts [7]. The process of averaging several acquisitions of the same image in order to enhance the quality has been used for years in magnetic resonance imaging, computed tomography and optical coherence tomography (OCT) [8–10]. For OCT B-scans, an averaging of the retina scans is integrated into the algorithm of most OCT devices, in order to reduce the speckle noise [11]. The same approach has been used by Sadda and other groups to perform multiple en face OCTA image averaging [7, 12–14], and its effectiveness on the overall quality of the image has been shown (reduction in background noise, smoother vessel walls and sharper contrast [13]). However, averaging can also modify the quantitative automatic measurements. For example, averaging can lead to a decrease in the vessel density (VD) of the superficial (SVP) and deep vascular plexuses [13]. Averaging can also decrease the flow void area and lead to an increase in the vascular area density of the choriocapillaris [15]. Likewise, using averaged images could hypothetically lead to the loss of visualisation of small retinal vessels, and therefore lead to a loss of information. Previous studies have shown the benefit of averaging on quantitative measurements, but no studies have qualitatively assessed the vessel segment integrity and no data in eyes with macular oedema are available.

The aim of this study was to analyse the effect of image averaging on the vessel detection using OCTA and to confirm the repeatability of automatic VD measurements in multiple images.

## Methods

### Participants

It was an observational case series study conducted in a tertiary ophthalmology centre (Lariboisière Hospital, Paris University, Paris, France). Eyes of patients with macular oedema and healthy controls were consecutively included over a 1-month period. Inclusion criteria for eyes with macular oedema were: patients aged ≥18 years with type 1 or 2 diabetes; presence of DME (defined as a retinal thickness >298 μm in the central subfield corresponding to the normal value plus 2 SDs: $260 + (2 \times 19)$ μm, and/or intra- or subretinal fluid seen on the OCT B-scan). Controls were healthy volunteers. Each image of the patients being compared to itself

and not to that of the controls, it did not seem necessary to us to match the control group on age.

This observational study was approved by the Ethics Committee of the French Society of Ophthalmology (IRB 00008855 Société Française d'Ophtalmologie), and a written informed consent was obtained from all patients to authorize the review of their records.

Inclusion criteria were healthy volunteers with no history of ocular disease for the control group and eyes with macular oedema, defined as the presence of intraretinal cystoid spaces in the central 3x3-mm image, and with a best-corrected visual acuity of at least 20/40. Exclusion criteria were the presence of media opacities and, for technical reasons, we did not include individuals with poor fixation capacities, since each OCTA image acquisition takes about 1 minute, and poor quality images would have been obtained.

## Image acquisition and averaging

All eyes were imaged using the OptoVue HD OCTA device (AngioVue model, manufactured by OptoVue, 2800 Bayview Drive, Fremont, CA 94538, USA) always by the same technician, and with a limited time between each acquisition. The OptoVue HD OCTA is part of routine care in eyes with macular oedema. A total of five sequential 3x3-mm and five sequential 6x6-mm scans centered on the fovea were obtained for each eye. The number of 5 sequential scans was chosen for the averaging because no significant difference in vessel length density (VLD) has been reported after averaging 5 images by Uji et al [13]. We used the terminology proposed by Campbell et al [16] to name the different retinal vascular plexuses. The preset parameters of the software were used to automatically segment the SVP and the deep vascular complex (DVC). The accuracy of the automatic segmentation was verified visually by scrolling the 320 B-scans.

The methodology for image averaging is shown in Fig 1. Briefly, all Jpeg formatted images of the SVP and DVC were extracted from the Optovue HD device and analysed using ImageJ software in its FIJI version. All the images were cropped in order to remove a square logo in the bottom left corner. As previously reported (Uji et al. [13]), averaging the raw images provided poor results because of eye motion between the acquisitions causing translation and rotational differences between the frames. To compensate for this misalignment, an image registration was performed: first, a linear registration to align the 5 images, and then an elastic registration to correct for deformations due to the spherical conformation of the eyes. Following a method described in [13], we stacked the 5 cropped images of each eye in each format and performed a linear registration using the StackReg plugin (bigwww.epfl.ch/thevenaz/stackreg, parameter = Scale rotation). We then performed an elastic registration using the first acquired image as a reference and the bUnwarpJ plugin (imagej.net/BUnwarpJ). The image averaging was performed using the "Average Intensity" process in FIJI, in the Z-projection mode. The process was then automated using FIJI macro.

## Image analysis

**Repeatability of automatic vessel density measurements in non-averaged images.** We first analysed the repeatability of automatic VD measurements obtained using AngioVue OCTA software in non-averaged images. All non-averaged images were analysed separately. The intra-class correlation coefficient (ICC) between the 5 automatic VD measurements obtained in a same area (3x3 or 6x6 mm) and a same slab (SVP, DVC or full retina slab) in one eye was calculated. After analysing the overall ICC results, the ICC results were compared between patients and controls, scan areas (3x3mm or 6x6mm), and slabs (SVP, DVC or full retina slab). We also analysed the correlation between the Signal Strength Index (SSI, a quality

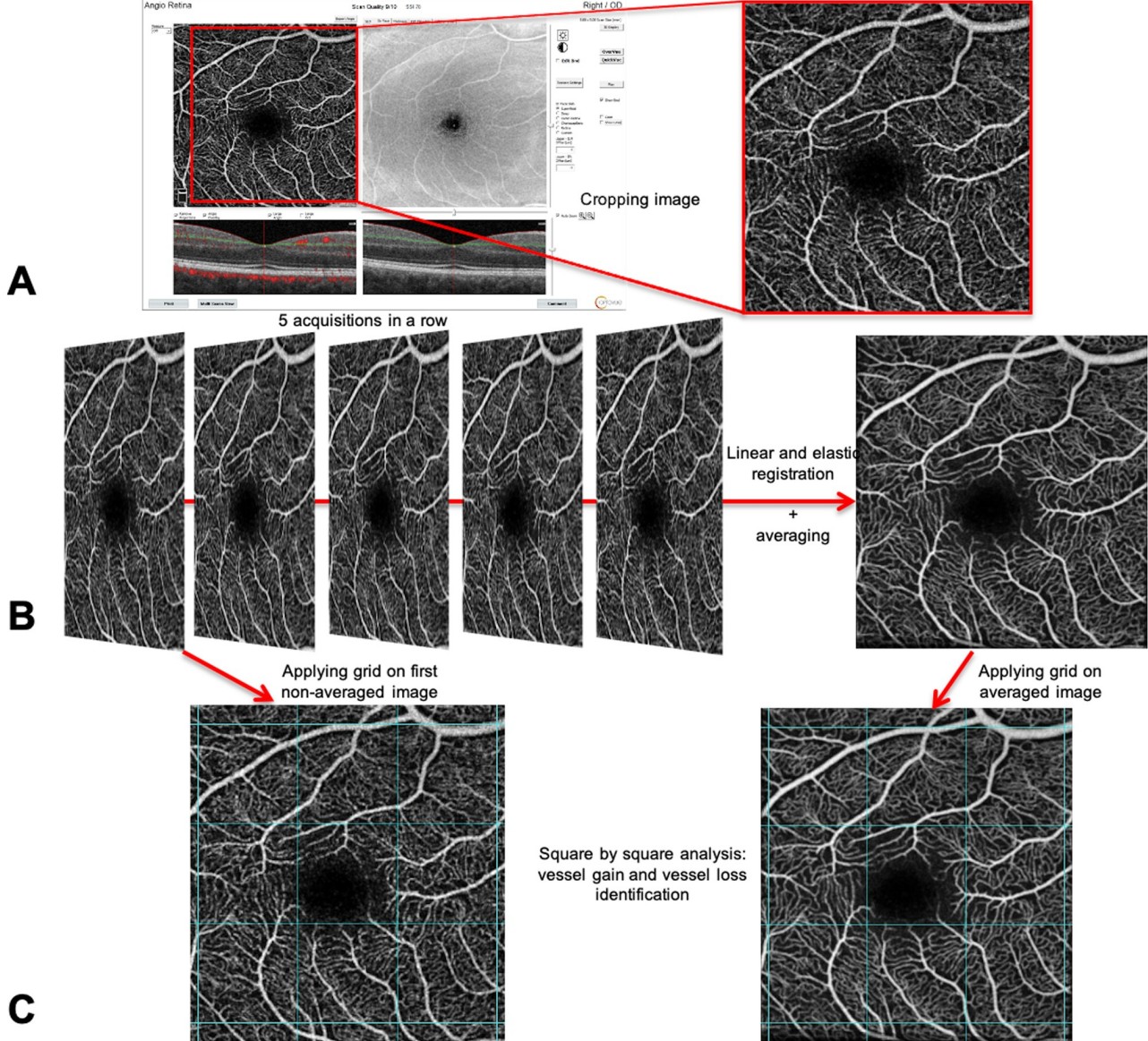

**Fig 1. Methodology flowchart for the averaging and qualitative analysis of non-averaged and averaged images. A.** All Jpeg formatted images were extracted from the Optovue HD device and analysed using ImageJ software in its FIJI version. All the images were cropped to remove the square logo in the bottom left corner. **B.** To compensate for the misalignment due to eye motion, an image registration was performed: first, a linear registration was performed to align the 5 images, and then an elastic registration was applied to correct for deformations due to the spherical conformation of the eyes. **C.** After alignment, all the first non-averaged images acquired and all averaged images were divided into 9 identical boxes composed of nine 172x172 pixel squares (composed of about 29,500 pixels), for semi-quantitative grading.

index given by the Optovue HD device for each image) and the automatic VD measurement in the SVP and DVC, using the Spearman correlation coefficient. All the statistical analyses were performed using R software.

**Impact of averaging on capillary detection.** The aim of this study was to assess the change in vessel detection between non-averaged and averaged OCTA images. The original non-averaged images and the averaged images were compared using both quantitative and qualitative approaches.

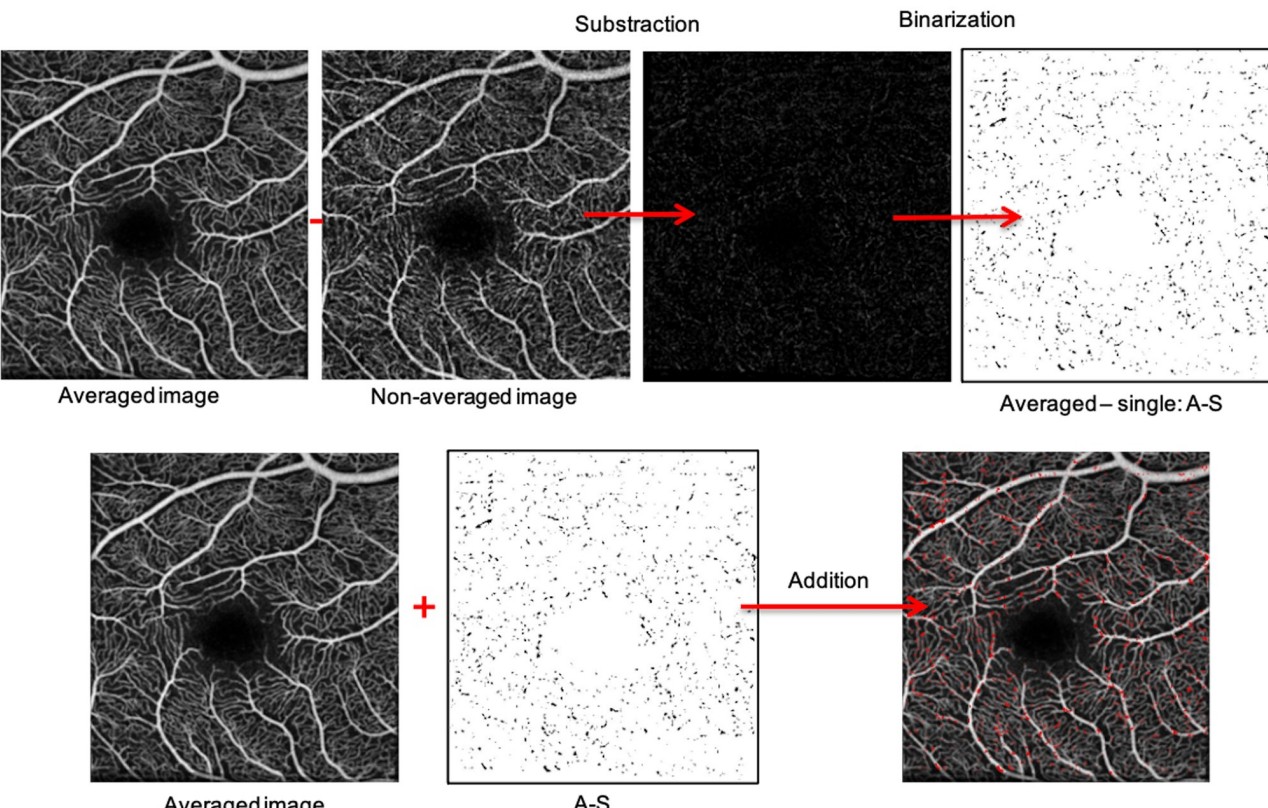

**Fig 2. Methodology flowchart for the quantitative analysis of non-averaged and averaged images.** We first constructed a pixel-by-pixel subtraction of a single non-averaged image (the first non-averaged image acquired) from the averaged image, that we called A-S (average–single) (first row). This A-S image was then binarized using a grey scale intensity threshold and colorized in red. Then, this red binarized A-S image was superimposed on the original averaged image, allowing a direct visualisation of the pixel gain (second row).

First, we performed a quantitative analysis of the change in capillary detection in the averaged images by comparing the pixel measurement between the single non-averaged images and the averaged images using ImageJ software. The methodology for this analysis is shown in Fig 2. We first constructed a pixel-by-pixel subtraction of a single non-averaged image (the first non-averaged image acquired) from the averaged image, that we called A-S (average–single). This A-S image was then binarized using a grey scale intensity threshold and colorized in red. Then, this red binarized A-S image was superimposed on the original averaged image, allowing a direct visualisation of the pixel gain (Fig 2).

The same process was then used to subtract the averaged image from the single non-averaged image. This image was called S-A, and the same binarization and colorization methods were used.

Regarding the threshold choice, it appeared that in the SVP images, the limit between the retinal vessels and the background was of about 40 on a scale of grey values from 0 to 255. In the DVC images, the threshold was however set to 60 to yield almost the same volume of significant area.

Lastly, the VD, defined as the ratio of the area occupied by vessels divided by the total area, was assessed in the SVP and in the DVC in the first non-averaged and averaged 3x3-mm images. For these VD measurements, all images were binarized using the Autothresholding

function in ImageJ. All the pixel areas are expressed in square pixels. The processes were automated using FIJI macro.

Secondly, a qualitative analysis of all the 3x3-mm OCTA images was performed, and the images were independently graded by two retina specialists blinded to image information (HL and MC). The capillary detection ability was compared between the first non-averaged image acquired and the averaged image. After alignment, all OCTA images were divided into 9 identical boxes composed of nine 172x172 pixel squares (composed of about 29,500 pixels), for semi-quantitative grading (Fig 1C). For a direct comparison between the non-averaged and averaged images, all boxes were placed next to each other. The presence of capillaries that could be detected only in the averaged image (but not detected in the non-averaged image) was assessed as a binary variable (graded as 1 if present or 0 if absent) in each box and was considered a "gain" in capillary detection. The number of boxes with a gain in capillary detection was counted for each averaged image and called the "gain score" of the image.

The presence of capillaries that could be detected only in the non-averaged image (but not in the averaged image) was also assessed as a binary variable (graded as 1 or 0) and was considered a loss in capillary detection. The number of boxes with a loss in capillary detection was counted for each averaged image and called the "loss score" of the image. The distribution of the averaged image gain and loss scores was compared using a t-test. The inter-reader correlation (using the Pearson correlation test for the gain, loss, and overall) and the rate of agreement (calculated as the number of identical answers divided by the total number of answers) were assessed.

## Results

### Participants

Ten eyes of 5 healthy volunteers and 10 eyes of 5 patients with macular oedema were included. Five acquisitions of 3x3mm and five acquisitions of 6x6mm centered on the fovea were performed in each eye. Thus, a total of $10_{(number\ of\ patients)}$ x $2_{(eyes)}$ x $5_{(acquisitions)}$ x $2_{(scan\ types)}$ = 200 acquisitions were performed. Among the 10 participants, there were 3 women (1 healthy volunteer and 2 patients). The mean age was 27.5 years [range: 25–29] for the healthy controls and 68.0 years [range: 66–72] for the patients. Macular oedema was secondary to diabetic retinopathy in 8 eyes and to age-related macular degeneration with type 1 choroidal neovascularisation in two eyes.

### Repeatability of automatic VD measurements in the non-averaged images

We obtained a total of $10_{(number\ of\ patients)}$ x $2_{(eyes)}$ x $2_{(scan\ areas)}$ x $3_{(slabs)}$ = 120 ICC results. The overall ICC analysis showed a good repeatability with a mean ICC value of $0.929 \pm 0.088$ and a median ICC value of 0.958. However, the ICC results significantly differed depending on the scan area, the presence of macular oedema, or the slab. Regarding the scan area, the ICC values were significantly higher when the 3x3mm acquisitions were used compared to the 6x6mm acquisitions (mean ICC: 0.963 *versus* 0.895; p < 0.001). The mean ICC values were also significantly different between patients and controls (0.892 and 0.965, respectively; p < 0.001). Regarding the slab, the repeatability was significantly higher in the SVP slab than in the DVC slab (mean: 0.967 *versus* 0.896; p = 0.001), while the use of the full retina slab showed intermediate results (mean: 0.924).

The SSI significantly and positively correlated with the automatic VD measurements, with Spearman correlation coefficients of 0.73 (p-value < 0.001) and 0.84 (p-value < 0.001) for the SVP and DVC slabs, respectively.

## Quantitative analysis of the impact of averaging on capillary detection

The pixel change (gain or loss) area after averaging was measured in the whole cohort, as well as in each group and each slab (Fig 3). In the whole cohort, the mean pixel gain area was of 790.4 ± 569.7 pixels/box and the mean pixel loss area was of 727.2 ± 394 pixels/box (p = 0.080). Interestingly, in the control eyes, the mean pixel gain area was significantly higher than the mean pixel loss area (996.1± 505.5 and 700.1 ± 374.7 pixels/box, respectively, p < 0.001), while in the eyes with macular oedema, the mean pixel gain area was significantly lower than the mean pixel loss area (584.4 ± 556.8 and 754.3 ± 411.6 pixels/box, respectively, p = 0.001).

The mean pixel gain was higher in the SVP compared to the DVP in the whole cohort (p = 0.002), and it was higher in control eyes compared to eyes with macular oedema (p < 0.001) (Table 1). Conversely, the mean pixel loss did not significantly differ between eyes with macular oedema and control eyes (p = 0.190), but was higher in the SVP of eyes with macular oedema compared to the SVP of control eyes (p < 0.001).

The superimposition of the red A-S images on the averaged images allowed assessing the location of the pixel gain and loss in the averaged images and showed that the gained pixels were located around the large vessels of the SVP or completing interruptions in the vessel walls (Figs 4 and 5). The pixel loss in the averaged images could either be due to the reduction in background noise and motion artifacts or to a real segment vessel loss, that was then assessed using a semi-quantitative analysis.

The mean VD in the whole cohort was significantly higher in the averaged images (12,962.1 pixels/box *versus* 12,559.8 pixels/box in the non-averaged images; p < 0.001). In eyes with macular oedema, the mean VD was 12,530 pixels/box in the averaged images *versus* 12,150 pixels/box in the non-averaged images (p < 0.001). In healthy control eyes, the mean VD was 13,400 pixels/box in the averaged images *versus* 12,970 pixels/box in the non-averaged images (p < 0.001). In the SVP, the mean VD was 12,380 pixels/box in the averaged images *versus* 12,220 pixels/box in the non-averaged images (p = 0.082). In the DVP, the mean VD was 13,550 pixels/box in the averaged images *versus* 12,900 pixels/box in the non-averaged images (p < 0.001).

## Qualitative analysis of the impact of averaging on capillary detection

A total of $10_{(number\ of\ patients)}$ x $2_{(eyes)}$ x $5_{(acquisitions)}$ x $2_{(scan\ areas)}$ x $2_{(slabs)}$ = 400 single non-averaged images and a total of $10_{(number\ of\ patients)}$ x $2_{(eyes)}$ x $2_{(scan\ areas)}$ x $2_{(slabs)}$ = 80 averaged images were obtained. The semi-quantitative grading showed that 99.6% (n = 358.5 out of 360) of boxes from the averaged images presented a gain in capillary detection, and none of the boxes presented a loss in capillary detection compared to the boxes of a single non-averaged image. Regarding the different slabs, we found that the SVP and DVC slabs had 99.2% (n = 178.5 out of 180) and 100% (n = 180 out of 180) of boxes presenting a gain in vessel detection, respectively (p = 0.683), while the SVP slab showed a higher number of boxes with a loss in pixel detection, i.e., a reduction in background noise or motion artifacts (43.9% [n = 79 out of 180] and 20.3% [n = 36.5 out of 180] for the SVP and DVC slabs, respectively, p < 0.001).

The overall inter-observer agreement was good with a correlation coefficient of 0.739, and a rate of agreement of 88.1%. However, when analysing separately the gain evaluation and the loss evaluation, we found that the inter-observer agreement was very good for the gain evaluation with a coefficient of 1 and a rate of agreement of 99.2%, while it was worse for the loss evaluation with a correlation coefficient of 0.489 and a rate of agreement of 76.9%. These results are summarized in Table 2.

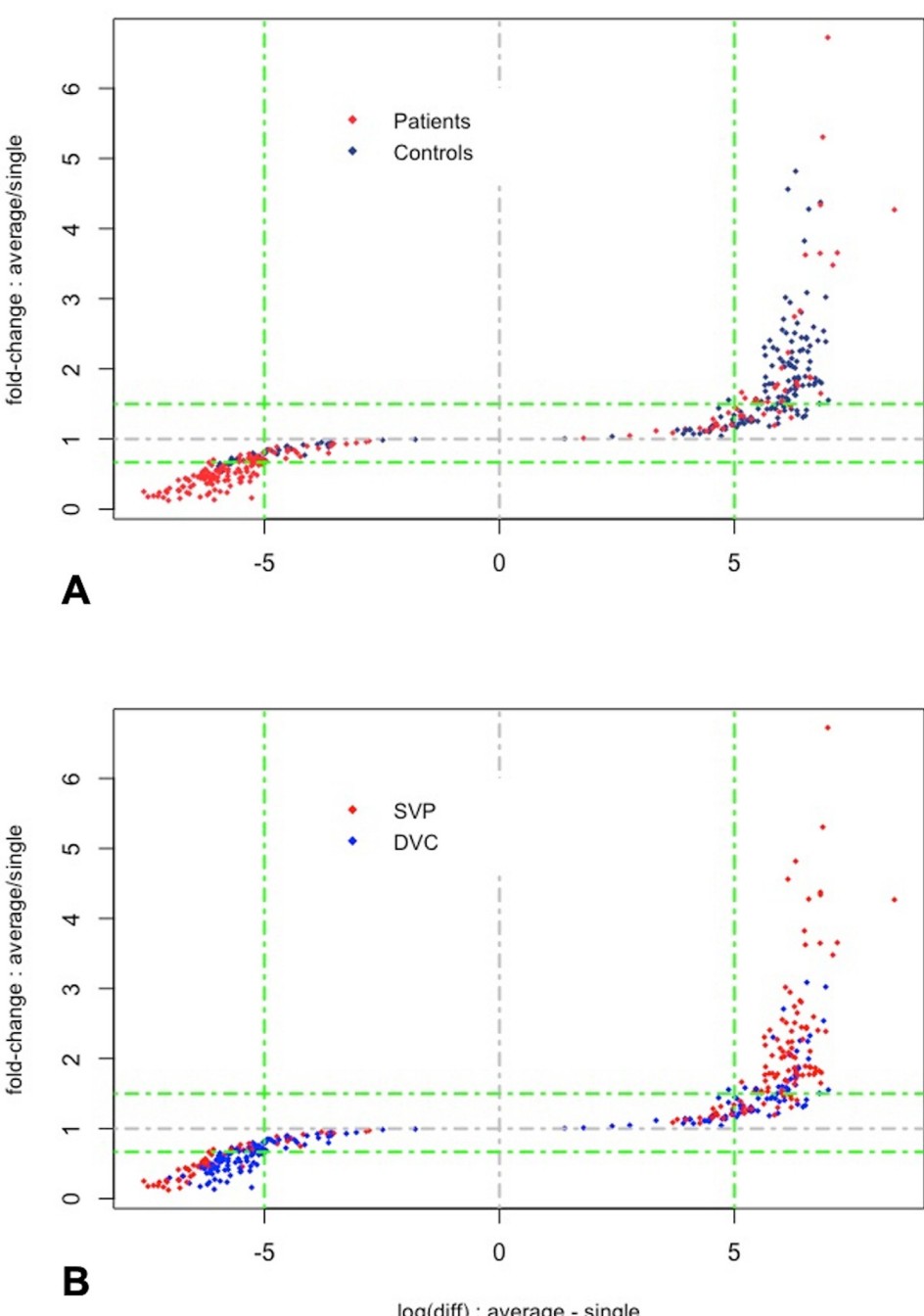

**Fig 3. Analysis of the differences in pixel areas between the averaged and single non-averaged images.** On the x-axis is represented the log(average-single), on the y-axis is represented the fold change (average/single). **A.** The pixel gain area was significantly higher than the pixel loss area in the control eyes (blue dots, mainly located in the upper right corner), while the pixel gain area was lower than the pixel loss area in eyes with macular oedema (red dots mainly located in the lower left corner). **B.** The pixel gain area was significantly higher than the pixel loss area in the SVP slab images (red dots, mainly located in the upper right corner), while the pixel gain area was lower than the pixel loss area in the DVP slab images (blue dots mainly located in the lower left corner).

**Table 1. Mean values ± standard deviation of pixel gain and loss in each group in averaged binarized images.**

|  | Whole cohort | Healthy control eyes | Eyes with macular oedema | P value |
|---|---|---|---|---|
| **Pixel GAIN** |  |  |  |  |
| Full retina slab | 790.4 ± 569.7 | 996.5 ± 505.5 | 584.4 ± 556.8 | < **0.001²†** |
| SVP slab | 884.0 ± 578.5 | 964.7 ± 410.8 | 803.2 ± 700.9 | 0.06† |
| DVP slab | 696.9 ± 546.5 | 1028.3 ± 585.8 | 365.5 ± 188.5 | < **0.001†** |
|  |  |  |  | **0.002‡** |
| **Pixel LOSS** |  |  |  |  |
| Full retina slab | 727.2 ± 394.0 | 700.1 ± 374.7 | 754.3 ± 411.6 | 0.19† |
| SVP slab | 743.0 ± 428.7 | 576.2 ± 275.1 | 909.9 ± 487.4 | < **0.001†** |
| DVP slab | 711.4 ± 356.4 | 824.0 ± 419.1 | 598.8 ±- 232.8 | < **0.001†** |
|  |  |  |  | 0.44‡ |

† Comparison of healthy eyes versus eyes with macular oedema.

‡ Comparison of the SVP versus the DVP in the whole cohort.

Bold values indicate a p-value <0.05.

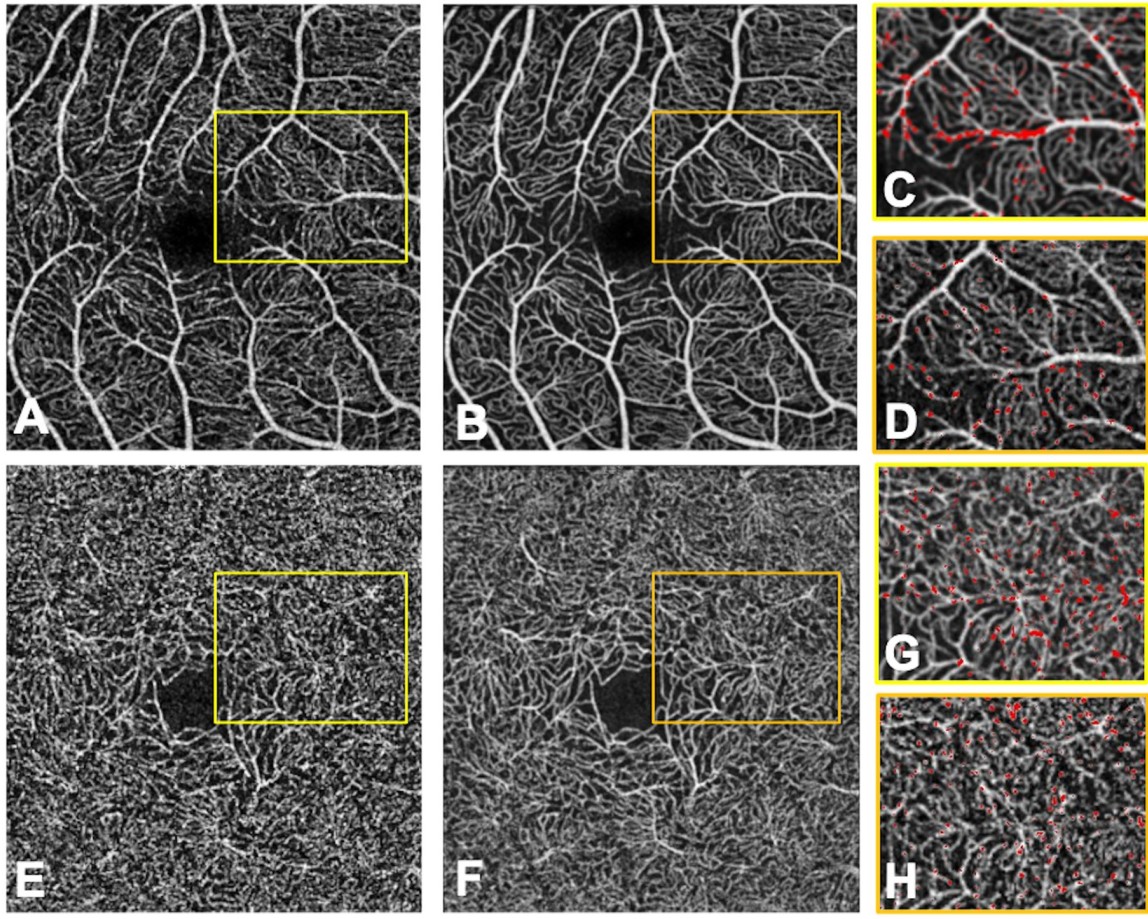

**Fig 4.** Single non-averaged (A, E) and averaged (B, F) 3x3mm OCTA images of the superficial (A, B) and deep (E, F) vascular plexuses of the right eye of a healthy control. The magnified views (C, G) of the averaged images with red pixels representing the A-S (averaged—single) binarized image highlight the "pixel gain". The magnified views (D, H) of the single non-averaged images with red pixels representing the S-A (single—averaged) binarized image highlight the "pixel loss".

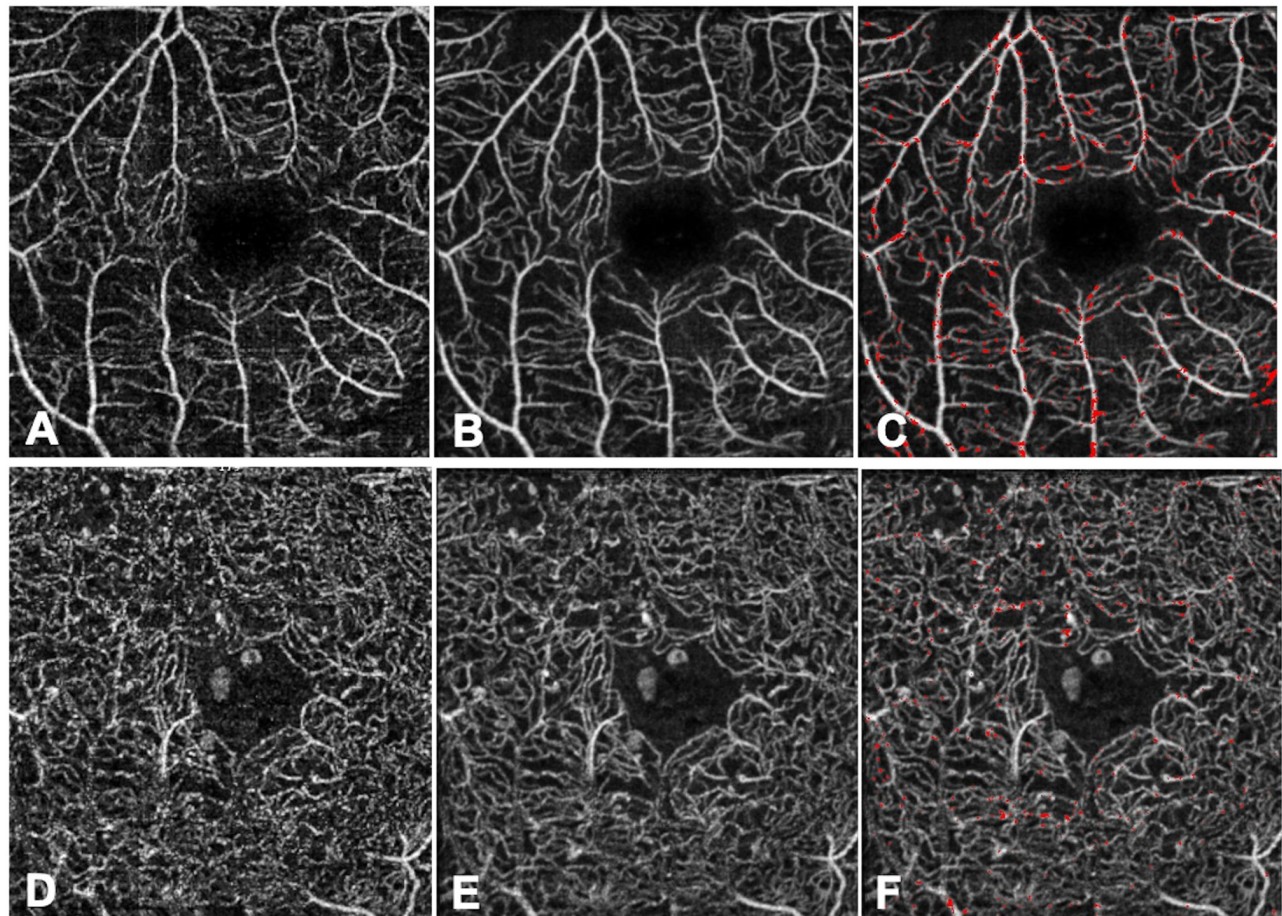

**Fig 5.** Single non-averaged (A, D) and averaged (B, E) 3x3mm OCTA images of the superficial (A, B, C) and deep (D, E, F) vascular plexuses of the right eye of a patient with diabetic macular oedema. The averaged images with red pixels (C, F) represent the A-S (averaged—single) binarized image and highlight the "pixel gain".

**Table 2. Inter-observer correlation and rate of agreement between the two readers for the qualitative assessment of pixel gain or loss in the averaged images compared to the single non-averaged images.**

|  | Observer 1 | Observer 2 | Pearson correlation coefficient | Agreement Rate (%) |
|---|---|---|---|---|
| **Full retina slab** |  |  |  |  |
| Gain | 0.992 | 1 | 1 | 99.2 |
| Loss | 0.369 | 0.272 | 0.489 | 76.9 |
| Overall | - | - | 0.739 | 88.1 |
| **SVP slab** |  |  |  |  |
| Gain | 0.983 | 1 | 1 | 98.3 |
| Loss | 0.456 | 0.422 | 0.528 | 76.7 |
| Overall | - | - | 0.693 | 87.5 |
| **DVP slab** |  |  |  |  |
| Gain | 1 | 1 | 1 | 100 |
| Loss | 0.283 | 0.122 | 0.368 | 77.2 |
| Overall | - | - | 0.775 | 88.6 |

The rate of agreement was calculated as the number of identical grading divided by the total number of grading.

## Discussion

In this study, we aimed to assess quantitatively the repeatability of automatic VD measurements and the impact of averaging on image quality and VD measures, but also to qualitatively and accurately analyse the impact of averaging on vessel and capillary segment visualisation. Our results confirmed that averaging allowed an overall enhancement of image quality by reducing the background noise and increasing vessel continuity, without any loss of visualisation of the vessel segments on the OCTA images both in healthy eyes and, in a greater proportion, in eyes with macular oedema.

First, we confirmed the good repeatability of automatic VD measurements using AngioVue software (mean overall ICC: 0.924) between 5 sequential non-averaged images. Interestingly, the repeatability was higher in healthy control eyes than in eyes with macular oedema (p < 0.001). This could be explained by the better fixation abilities of healthy eyes. The repeatability of automatic VD measurements was also higher in the SVP slab compared to the DVC slab (p = 0.001), probably due to the more complex organisation of the capillaries in the DVC, preventing a clear distinction between the capillaries and the background noise [6, 17, 18]. This variability in VD measurements in the DVC could also be explained by the physiological spatio-temporal heterogeneity of the macular perfusion that could be detected in these 5 sequential acquisitions [19].

We then assessed the impact of averaging on vessel segment detection using quantitative and qualitative approaches. Previous studies have shown the benefit of averaging on quantitative measurements such as VD and VLD [7, 12, 13, 20]. In these studies, averaging has allowed to reduce the influence of the background noise due to an artifactual flow signal and to improve vessel delineation. However, no studies have specifically focused on eyes with macular oedema and assessed a potential loss of vessel segments due to averaging. Our quantitative analysis aimed at measuring and locating the pixel gain and loss between the non-averaged and averaged images. The mean pixel loss was higher in eyes with macular oedema compared to healthy eyes and this pixel loss was due to the reduced background noise without any real disappearance of vessel segments. Our semi-quantitative analysis performed in 9 boxes in each eye showed that about a third of the boxes showed an improvement in quality with a pixel loss corresponding to the reduction in background noise. This pixel loss in the averaged images was less frequent in the DVC slab than in the SVP slab, probably because of the complexity of the DVC with a stronger background noise and the presence of anastomotic vessels. By adapting to the neuronal metabolism, some retinal capillaries could not be constantly circulating, which could have led to a loss of vessel segment detection after averaging [19]. However, in our study, the qualitative analysis of all areas of pixel loss confirmed that averaging did not lead to any loss of vessel detection. Averaging was even more beneficial in eyes with macular edema, since the presence of intra-retinal fluid could impair capillary detection and increase background noise. Our results confirmed that averaging has allowed to enhance image quality and reduce the background noise in these eyes.

Our study has some limitations. The first limitation is the small sample size and the younger age of the healthy volunteers compared to patients with macular oedema, as it has been shown that the retinal vascular density decreases with age [21]. However, the main objective of the study was not to compare patients and controls but to assess the intra-individual repeatability and averaging capacities. Regarding the quantitative analysis of the impact of image averaging, the main limitation was related to the choice of the contrast thresholds. Indeed, the choice of the binarization thresholds has been shown to provide different results in the image analysis [22], and could be a major limitation in the interpretation of quantitative automatic measurements.

To conclude, using sequential OCTA images, we confirmed the good repeatability of automatic VD measurements. Our results confirmed the positive impact of averaging OCTA images on the image analysis as well as on the background noise detection.

## Supporting information

**S1 Data.**
(XLSX)

## Author Contributions

**Conceptualization:** Ramin Tadayoni, Aude Couturier.

**Data curation:** Hugo Le Boité, Mardoche Chetrit, Sophie Bonnin, Aude Couturier.

**Formal analysis:** Hugo Le Boité, Mardoche Chetrit, Ali Erginay, Sophie Bonnin, Carlo Lavia.

**Investigation:** Ramin Tadayoni.

**Methodology:** Hugo Le Boité, Ali Erginay, Sophie Bonnin, Carlo Lavia.

**Project administration:** Aude Couturier.

**Supervision:** Mardoche Chetrit, Ali Erginay, Carlo Lavia, Aude Couturier.

**Validation:** Ali Erginay, Sophie Bonnin, Carlo Lavia, Ramin Tadayoni.

**Visualization:** Sophie Bonnin.

**Writing – original draft:** Hugo Le Boité, Mardoche Chetrit.

**Writing – review & editing:** Ramin Tadayoni, Aude Couturier.

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
