## [Decision Letter · Decision Letter 0]

11 Jun 2021

PONE-D-21-17138

Impact of Image Averaging on Vessel Detection using Optical Coherence Tomography Angiography in Eyes with Macular Oedema and in Healthy Eyes

PLOS ONE

Dear Dr. Couturier,

Thank you for submitting your manuscript to PLOS ONE. After careful consideration, we feel that it has merit but does not fully meet PLOS ONE’s publication criteria as it currently stands. Therefore, we invite you to submit a revised version of the manuscript that addresses the points raised during the review process.

We look forward to receiving your revised manuscript.

Kind regards,

Demetrios G. Vavvas

Academic Editor

PLOS ONE

Journal Requirements:

3. In your Methods section, please clarify whether imaging of eyes using the OptoVue HD OCTA device was conducted as part of routine care, or for the purposes of research.

"I have read the journal's policy and the authors of this manuscript have the following

competing interests:

Dr. LE BOITE, Dr. CHETRIT, Dr ERGINAY and Dr LAVIA have nothing to disclose.

Dr BONNIN reports personal fees from Allergan, outside of the submitted work.

Dr. COUTURIER reports personal fees from Allergan, Bayer, Horus and Novartis,

outside of the submitted work.

Dr. TADAYONI reports personal fees from Novartis, Bayer, Roche, Genentech,

Allergan, Zeiss, Alcon and Oculis, outside of the submitted work.

All authors attest that they meet the current ICMJE criteria for authorship."

Reviewers' comments:

Reviewer's Responses to Questions

**Comments to the Author**

1. Is the manuscript technically sound, and do the data support the conclusions?

Reviewer #1: Yes

Reviewer #2: Partly

2. Has the statistical analysis been performed appropriately and rigorously? 

Reviewer #1: Yes

Reviewer #2: Yes

3. Have the authors made all data underlying the findings in their manuscript fully available?

Reviewer #1: No

Reviewer #2: Yes

4. Is the manuscript presented in an intelligible fashion and written in standard English?

Reviewer #1: Yes

Reviewer #2: Yes

5. Review Comments to the Author

Reviewer #1: Overall, the manuscript is well-written. The authors discuss the impact of image averaging in OCTA and in eyes with macular edema.

Please find attached the comments:-

1. Introduction is good but there is a paragraph missing describing the gap in literature, or why is the study being done. It would be good to add this to for the flow.

2. Please describe the inclusion criteria for inclusion of macular edema patients. Were there any specific criteria or what group of macular edema patients were recruited?

3. Please be consistent with decimal places. I think p value should be reported upto 3 places for line 43, 44, 230, 232, 234, 236, 250, 251, 298, 300, 302, 303, 343, 345

4. Table 1 should also include the SD along with mean

5. The discussion and conclusions in abstract should also mention about the significance of studying the effect of image averaging in macular edema patients, as that seems to be the main novel idea of the project.

Reviewer #2: Some linguistic parameters should be addressed, and additional analysis over some of the conclusions would be helpful. The effect of normal aging on vascular density should also be mentioned, due to the study of non age-matched controls. Please find more information in the pdf attached.

6. PLOS authors have the option to publish the peer review history of their article (what does this mean?). If published, this will include your full peer review and any attached files.

Reviewer #1: No

Reviewer #2: No

---

## [Author Response · Author response to Decision Letter 0]

8 Sep 2021

We thank the reviewers for their positive feedback and constructive comments regarding our manuscript entitled “Impact of Image Averaging on Vessel Detection using Optical Coherence Tomography Angiography in Eyes with Macular Oedema and in Healthy Eyes”.

We have been able to provide all the information requested, and we have taken into account the comments to improve our manuscript. 

Answer to Reviewer #1:

1. Introduction is good but there is a paragraph missing describing the gap in literature, or why is the study being done. It would be good to add this to for the flow.

We thank the reviewer for his remark and agree. We have added the following paragraph, as suggested:

“Previous studies have shown the benefit of averaging on quantitative measurements, but no studies have qualitatively assessed the vessel segment integrity and no data in eyes with macular oedema are available.”

2. Please describe the inclusion criteria for inclusion of macular edema patients. Were there any specific criteria or what group of macular edema patients were recruited?

The inclusion criteria for the macular oedema patients have been added:

“Inclusion criteria for eyes with macular oedema were: patients aged ≥18 years with type 1 or 2 diabetes; presence of DME (defined as a retinal thickness >298 µm in the central subfield corresponding to the normal value plus 2 SDs: 260 + (2 × 19) μm, and/or intra- or subretinal fluid seen on the OCT B-scan).”

3. Please be consistent with decimal places. I think p value should be reported upto 3 places for line 43, 44, 230, 232, 234, 236, 250, 251, 298, 300, 302, 303, 343, 345

We have changed the p values with 3 decimals in all the manuscript as suggested.

4. Table 1 should also include the SD along with mean

We have added the SD along with mean in Table 1 as suggested.

5. The discussion and conclusions in abstract should also mention about the significance of studying the effect of image averaging in macular edema patients, as that seems to be the main novel idea of the project.

The thank the reviewer for his remark and agree. 

We have mentioned in the discussion section line 420 that “The mean pixel loss was higher in eyes with macular oedema compared to healthy eyes and this pixel loss was due to the reduced background noise” 

In addition, we have added the following paragraph in the discussion section and in the abstract:

“Averaging is even more beneficial in eyes with macular edema, since the presence of intra-retinal fluid can impair capillary detection and increase background noise. Our results confirmed that averaging allow to enhance image quality and reduce the background noise in these eyes.”

Answer to Reviewer #2:

ABSTRACT 

Original text line: 36-40 Proposed changes: Please consider rephrasing this part in two separate, shorter sentences. 

We thank the reviewer for his suggestion. We have rephrasing the sentences as follows:

“The effect of the averaging of the 5 acquisitions on vessel detection was analysed quantitatively using a pixel-by-pixel automated analysis. In addition, two independent retina experts qualitatively assessed the change in vessel detection in averaged images segmented in 9 boxes and compared to the first non-averaged image”. 

Original text line: 42 Proposed changes: Please define intra-class correlation coefficient, before using the abbreviation ICC. 

We thank the reviewer for his comment. We have added the term “intra-class correlation coefficient” to define ICC, as suggested.

INTRODUCTION 

Original text line: 58 Proposed changes: Please avoid using two gerunds one after the other; for ex. you could replace assessing by the noun “assessment” 

We thank the reviewer for his suggestion. We have replaced the term by “assessment” as proposed.

Original text line: 66 Proposed changes: “This process of averaging..” 

We thank the reviewer for his suggestion. We have rephrased by “this process of averaging” as proposed.

Original text line: 66-70 Proposed changes: Please consider using the letter “the” less frequently; for ex. the quality, the speckle noise. Please consider this comment throughout the manuscript’s text.

We thank the reviewer for his remark. We have used the letter “the” less frequently as advised. 

Original text line: 73-77 Proposed changes: Please consider rephrasing this part.

We have rephrased the part as suggested:

“However, averaging can also modify the quantitative automatic measurements. For example, averaging can lead to a decrease in vessel density (VD) of the superficial (SVP) and deep vascular plexuses(13). Averaging can also decrease the flow void area and lead to an increase in the vascular area density of the choriocapillaris (15).”

Original text line: 79 Proposed changes: “..lead to loss of information”. 

We thank the reviewer for his remark. We have made the proposed change.

RESULTS 

Original text line: 218-219 Proposed changes: The authors should probably justify the use of non age-matched controls for the purpose of their study (in one sentence). 

We thank the reviewer for his comment. Each image of the patients being compared to itself and not to that of the controls, it did not seem necessary to us to match the control group on age. 

We have added the following sentences to the methods section lines 159-161:

“Controls were healthy volunteers. Each image of the patients being compared to itself and not to that of the controls, it did not seem necessary to us to match the control group on age.

The effect of normal aging on vascular density has been mentioned as a limitation in the discussion section lines 434-435

DISCUSSION 

Original text line: 351-355 Proposed changes: Please consider rephrasing this part in two separate, shorter sentences.

We have rephrased this part as proposed:

“Previous studies have shown the benefit of averaging on quantitative measurements such as VD and VLD. In these studies, averaging has allowed to reduce the influence of the background noise due to an artifactual flow signal and to improve vessel delineation.”

Answers to the Journal Requirements:

1. The manuscript meets PLOS ONE's style requirements, including those for file naming. 

2. The reference list to ensure that it is complete and correct 

3. In your Methods section, please clarify whether imaging of eyes using the OptoVue HD OCTA device was conducted as part of routine care, or for the purposes of research.

The OptoVue HD OCTA is part of routine care in eyes with macular oedema. This has been added to the Methods section.

4. We note that you have indicated that data from this study are available upon request. PLOS only allows data to be available upon request if there are legal or ethical restrictions on sharing data publicly. 

There are no restrictions, we have uploaded the minimal anonymized data set necessary to replicate the study findings. 

"I have read the journal's policy and the authors of this manuscript have the following

competing interests:

Dr. LE BOITE, Dr. CHETRIT, Dr ERGINAY and Dr LAVIA have nothing to disclose.

Dr BONNIN reports personal fees from Allergan, outside of the submitted work.

Dr. COUTURIER reports personal fees from Allergan, Bayer, Horus and Novartis,

outside of the submitted work.

Dr. TADAYONI reports personal fees from Novartis, Bayer, Roche, Genentech,

Allergan, Zeiss, Alcon and Oculis, outside of the submitted work.

All authors attest that they meet the current ICMJE criteria for authorship."

We confirm that the Competing Interests mentioned do not alter our adherence to all PLOS ONE policies on sharing data and materials

We have included the following statement: "This does not alter our adherence to PLOS ONE policies on sharing data and materials.”

---

## [Editor Report · Decision Letter 1]

13 Sep 2021

Impact of image averaging on vessel detection using optical coherence tomography angiography in eyes with macular oedema and in healthy eyes

PONE-D-21-17138R1

Dear Dr. Couturier,

We’re pleased to inform you that your manuscript has been judged scientifically suitable for publication and will be formally accepted for publication once it meets all outstanding technical requirements.

Kind regards,

Demetrios G. Vavvas

Academic Editor

PLOS ONE
---

## [Editor Report · Acceptance letter]

13 Oct 2021

PONE-D-21-17138R1 

Impact of image averaging on vessel detection using optical coherence tomography angiography in eyes with macular oedema and in healthy eyes 

Dear Dr. Couturier:

I'm pleased to inform you that your manuscript has been deemed suitable for publication in PLOS ONE. Congratulations! Your manuscript is now with our production department. 

Kind regards, 

on behalf of

Prof. Demetrios G. Vavvas 

Academic Editor

PLOS ONE